# GeRA: Label-Efficient Geometrically Regularized Alignment

## Abstract

Pretrained unimodal encoders incorporate rich semantic information into embedding space structures. To be similarly informative, multi-modal encoders typically require massive amounts of paired data for alignment and training. We introduce a semi-supervised **Ge**ometrically **R**egularized **A**lignment (GeRA) method to align the embedding spaces of pretrained unimodal encoders in a label-efficient way. Our method leverages the manifold geometry of unpaired (unlabeled) data to improve alignment performance. To prevent distortions to local geometry during the alignment process —potentially disrupting semantic neighborhood structures and causing misalignment of unobserved pairs — we introduce a geometric loss term. This term is built upon a diffusion operator that captures the local manifold geometry of the unimodal pretrained encoders. GeRA is modality-agnostic and thus can be used to align pretrained encoders from any data modalities. We provide empirical evidence to the effectiveness of our method in the domains of speech-text and image-text alignment. Our experiments demonstrate significant improvement in alignment quality compared to a variaty of leading baselines, especially with a small amount of paired data, using our proposed geometric regularization.

## 1 Introduction

Data comes in many modalities, including text, speech, images, and video. Unimodal encoders aim to extract the intrinsic features of data drawn from a single modality, representing it in an embedding space. The goal of multi-modal learning is to learn a *shared* representation space for encoders of different modalities. In this setting, objects captured in different modalities have common representations in this shared space. This task is commonly referred to as *multi-modal alignment* (Baltrušaitis et al., 2018). Finding unified representations unlocks applications that require multiple modalities, like retrieving and generating descriptions of visual content.

In this paper, we consider multi-modal alignment using pretrained unimodal encoders. We are given paired and unpaired multi-modal data of potentially different dimensionalities and aim to learn an alignment transformation into a common embedding space. Although the domain of image and text alignment has been extensively explored thanks to large, publicly available image-text datasets (Schuhmann et al., 2021), one quickly runs into data availability problems when looking at new modalities. Indeed, for most modality pairs, such as speech and text or protein sequences and biomedical texts (Xu et al., 2023), there are far fewer paired data points than for images and text.

With the scenario above in mind, we present a robust and data-efficient alignment method that generalizes to new modalities, even under limited paired data availability. Our key idea is to preserve the local geometric structure learned by the pretrained encoders (Moschella et al., 2023; Antonello et al., 2021). These geometric structures, however, are not explicitly leveraged by existing alignment methods. Specifically, learning an alignment using only a contrastive objective, as explored by Radford et al. (2021) and others, seemingly does not maintain the manifold geometry (see Figure 1) and requires substantial paired data for alignment. Conversely, the Procrustes method (Gower, 1975) aligns the datasets through an isometric rotation transformation and hence fully preserves the geometric structure. However, Procrustes has low plasticity.

Our proposed **Ge**ometrically **R**egularized **A**lignment (GeRA) method leverages semantically rich manifold structures and preserves local geometry, while allowing enough flexibility to learn a meaningful alignment. We use a regularization loss which optimizes for local geometry preservation,

built on a diffusion operator to capture the local geometry. We freeze the unimodal encoders during the alignment process, reducing computational costs. Our approach falls into the regime of semi-supervised learning, as we can leverage the vast amount of unpaired (unlabeled) data with relatively few pairs to establish alignment. See Figure 2 for an overview of our method.

**Contributions.** Our work advances the field of data-efficient multi-modal alignment by addressing several limitations of existing methods. Our main contributions are three-fold:

- **Geometry-Preserving Alignment:** We introduce a semi-supervised alignment method that aligns multi-modal data distributions while preserving local geometry. It exhibits both global flexibility to align the paired points and local geometric preservation to incorporate the rich semantic information of the manifold structure.
- **Efficiency:** A key advantage of the proposed method is its label efficiency, as it employs a semi-supervised approach to use unlabeled data. This enables the alignment to capture additional information from the pretrained unimodal encoders in regions where there are no labeled pairs.
- **Modality-Agnostic Formulation:** GeRA is agnostic to the choice of encoders and modalities; it does not rely on domain-specific knowledge like augmentation. We experiment across multiple encoders and data modalities to show that our method is effective across configurations. It can be efficiently applied whenever pretrained models are available.

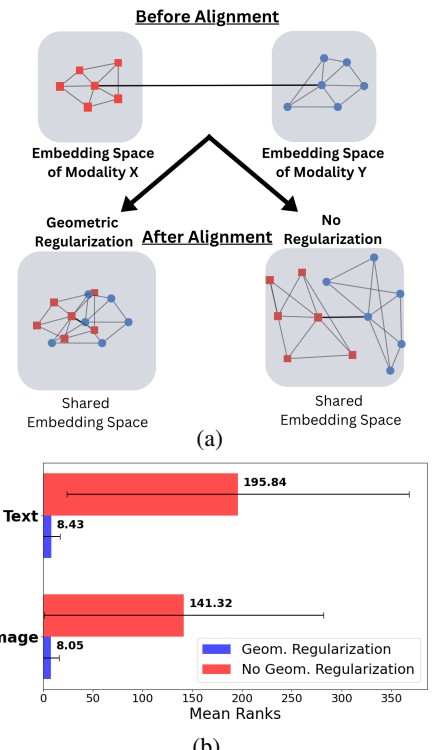

(a)

(b)

Figure 1: (a) Illustration of the effect of GeRA on alignment quality; GeRA preserves local neighborhoods, whereas non-regularized methods might distort them. Inter-modality black lines denote known pairs and gray lines denote neighbors. (b) Average ranking of the five nearest neighbors (before alignment) in the learned aligned spaces, using contrastive loss with and without our geometric regularization.

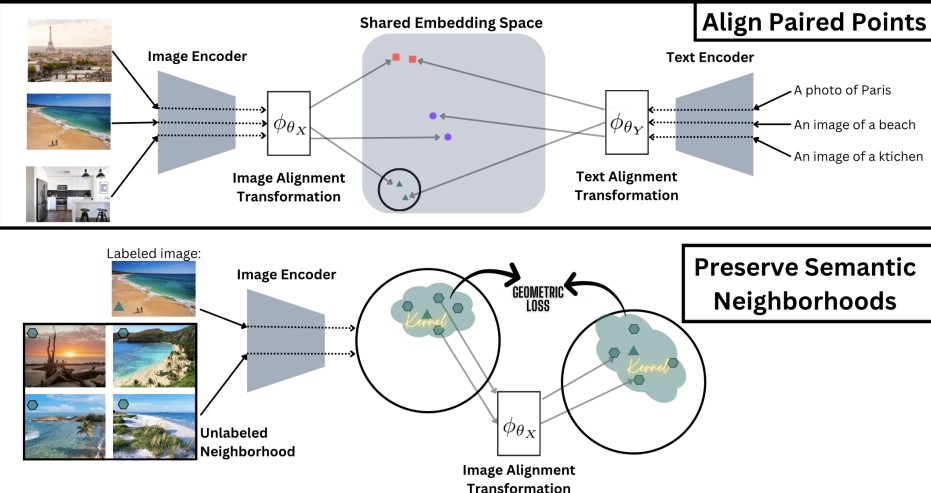

Figure 2: GeRA Training Approach: We optimize image and text alignment functions, focusing on achieving both global alignment of paired points and the preservation of local geometric structures.

## 2 RELATED WORK

Various multi-modal alignment methods have been introduced, each based on different assumptions on data availability and computational needs; most have been applied to text and image modalities.

**Training Multi-Modal Encoders:** Radford et al. (2021); Chen et al. (2022); Jia et al. (2021) jointly train image and text encoders from scratch, learning a shared representation for both modalities using a contrastive objective (Wang & Isola, 2020). This approach outperforms many existing models (Kolesnikov et al., 2020; Chen et al., 2020; He et al., 2016) in zero-shot classification on ImageNet (Deng et al., 2009). These methods, however, demand large training datasets (Gadre et al., 2023; Thomee et al., 2016; Sun et al., 2017) and consume significant computational resources. Zhai et al. (2021) reduce computational costs by freezing the image encoder. Due to the large training datasets (Sun et al., 2017; Thomee et al., 2016) this method remains computationally intensive.

**Relative and Anchor-based Encodings:** Moschella et al. (2023); Antonello et al. (2021) demonstrate that high-quality encoders produce semantically rich and consistent manifold structures. This observation suggests the concept of relative encodings, where a sample is encoded based on its neighborhood. Such relative encodings have been shown to remain consistent across various encoders and modalities (Moschella et al., 2023). Building on this idea, Norelli et al. (2022) ensures consistent encodings across different modalities using frozen pretrained encoders (Song et al., 2020; Dosovitskiy et al., 2020), eliminating the need for a training phase. Their method achieves high performance, coming close to models trained with substantially more data. However, a trade-off arises: inference time increases with the number of anchor points (labeled data) used.

**Unsupervised Alignment Techniques:** There has been also research efforts towards unsupervised alignment of embedding spaces without relying on paired modality data. The study Alvarez-Melis & Jaakkola (2018) uses the Gromov-Wasserstein optimal transport objective (Nekrashevich et al., 2023) to align word embeddings from various languages. Despite its advantage of not requiring labeled data, the method poorly scales to the 4th power in terms of the number of embedding points.

**Manifold Geometry:** Early works in manifold learning, such as Locally Linear Embedding (LLE) (Roweis & Saul, 2000), Isomap (Tenenbaum et al., 2000), and multi-dimensional scaling (Saeed et al., 2018), capture geometric properties of data manifolds while mapping them into simpler spaces. These methods leverage the rich local structure of datasets, constructing a lower-dimensional embedding that retains the topological and geometric characteristics of local neighborhoods in high-dimensional data space. In the context of semi-supervised learning, Sindhwani et al. (2005); Zhu et al. (2003) propose frameworks for integrating geometry learned from both labeled and unlabeled data into classification algorithms based on the graph Laplacian (Sindhwani et al., 2005), and based on a Gaussian random field model (Zhu et al., 2003). These works, however, focus on a unimodal setting and do not address semi-supervised alignment of data from multiple modalities.

## 3 PROBLEM FORMULATION

### 3.1 MULTI-MODAL ALIGNMENT

Consider two datasets $X \in \mathbb{R}^{N_X \times d_X}$ and $Y \in \mathbb{R}^{N_Y \times d_Y}$, originating from two distinct modalities. Here, $N_X$ and $N_Y$ denote the number of samples in each dataset, and $d_X$ and $d_Y$ represent the dimensions of $X$ and $Y$, respectively, which may differ. These datasets denote points embedded by pretrained unimodal encoders. We assume pairwise correspondence for only a small subset of these points, denoted by $\{(x_{p_i}, y_{q_i})\}_{i=1}^M$, where $x_{p_i} \in X$, $y_{q_i} \in Y$, $p_i \in [1, N_X]$ and $q_i \in [1, N_Y]$ are some index permutations, and $M \ll N_X, N_Y$. Those pairings correspond to the same objects captured by different modalities. All other points in $X$ and $Y$ are unpaired (unlabeled).

The task at hand is to align the data distributions into a common embedding space. While most methods focus only on aligning the paired data points, we propose to leverage unlabeled (unpaired) points from each modality to preserve the rich geometric structure of their original embedding spaces.

To approach this problem, we define two trainable alignment functions, namely $\phi_{\theta_X} : \mathbb{R}^{d_X} \to \mathbb{R}^d$ and $\phi_{\theta_Y} : \mathbb{R}^{d_Y} \to \mathbb{R}^d$, where $d$ is the dimension of the joint embedding space. These alignment transformations are modeled as neural networks. This approach is *encoder and modality agnostic*, requiring some pretrained unimodal encoder for each modality and a small paired dataset.

### 3.2 Preserving Manifold Geometry

Unimodal encoders, trained on large and often self-supervised datasets, learn to encode the data into a rich representation that accurately reflects the intrinsic structure of the data. When training the alignment using a smaller paired dataset, this manifold structure might be distorted, degrading the quality of the alignment. Existing methods are not required to preserve these structures, leaving a potential source of information unused. Specifically, learning an alignment using only a contrastive objective, as explored by (Radford et al., 2021), distorts the neighborhood geometry (see Figure 1b) and requires substantial paired data to learn an effective alignment.

We propose a geometrically regularized alignment that aligns the paired points while preserving local neighborhood structures, which is motivated by the relation between local neighborhoods and the (Riemannian) manifold geometry (e.g. in approximating geodesic distances) (Coifman & Lafon, 2006; Li & Dunson, 2019). This approach offers global flexibility for obtaining a meaningful alignment and local regularization to maintain the neighborhood structure. The geometry regularization pursues an intuitive goal of keeping similar objects close in the aligned space. This allows for more effective generalization of the learned alignment to nearby (unpaired) points, as evident from the improved performance by our approach (see Section 5.3.) For preserving local neighborhoods, unlabeled data can be leveraged, thus allowing a semi-supervised approach.

## 4 Geometrically Regularized Alignment Method

### 4.1 GeRA Loss Function

We introduce the GeRA loss, which optimizes for both aligning paired points and preserving the neighborhood structure of nearby unpaired points. This loss is semi-supervised, as it uses paired data for the alignment and captures the local geometry using both paired and unpaired data. The loss is defined as follows:

$$\mathcal{L}_{GeRA}(\theta_X, \theta_Y) = \mathbb{E}_{(X_B, Y_B) \sim P_{\text{Pos}}} \Big[ \underbrace{\mathcal{L}_{Con}(X_B, Y_B; \theta_X, \theta_Y) + \mathcal{L}_{Con}(Y_B, X_B; \theta_Y, \theta_X)}_{\text{Alignment}}$$
$$+ \underbrace{\alpha \cdot \big( \mathcal{L}_{Geo}(X_B; \theta_X) + \mathcal{L}_{Geo}(Y_B; \theta_Y) \big)}_{\text{Geometric Regularization}} \Big] \quad (1)$$

where $\theta_X$ and $\theta_Y$ parameterize the alignment transformations, $\phi_{\theta_X}$ and $\phi_{\theta_Y}$, respectively, $P_{Pos}$ represents the uniform distribution over all paired points from both modalities, and $B$ represents the number of paired data points in a batch.

**Alignment:** We align the labeled points via a contrastive loss, denoted by $\mathcal{L}_{Con}(X_B, Y_B; \theta_X, \theta_Y)$ as proposed by Radford et al. (2021). It minimizes the distance between positive pairs (paired points) while maximizing the distance of negative samples. We apply this loss to the alignment transformation outputs:

$$\mathcal{L}_{Con}(X_B, Y_B; \theta_X, \theta_Y) = -\frac{1}{2} \sum_{x \in X_B} \log \frac{\exp\big(\text{cossim}(\phi_{\theta_X}(x), \phi_{\theta_Y}(y))/t\big)}{\sum_{y \in Y_B} \exp\big(\text{cossim}(\phi_{\theta_X}(x), \phi_{\theta_Y}(y))/t\big)} \quad (2)$$

where $t$ is a temperature hyperparameter.

**Geometric Regularization:** Our geometric loss term aims to preserve the local geometric structure:

$$\mathcal{L}_{Geo}(X_B; \theta_X) = \frac{1}{B} \sum_{x \in X_B} \mathbb{E}_{N_K(x) \sim S(x)} \Big[ \big\| \mathbf{W}_{N_K(x)} - \mathbf{W}_{\phi_{\theta_X}(N_K(x))} \big\|_F^2 \Big]. \quad (3)$$

where $\mathbf{W}_{N_K(x)}$ and $\mathbf{W}_{\phi_{\theta_X}(N_K(x))}$ are some matrices encoding the neighborhood structure of sample $x$, and $N_K(x)$ denotes a sampled set of $K$ neighbors of $x$ (according to the original embedded space). This loss operates only within a single modality and is independent of the other modality. For a given batch of unimodal samples $X_B$, we sample a set of $K$ neighbors $N_K(x)$ (defined based on proximity in the original space) for each sample in the batch, drawn from a precomputed larger neighborhood distribution $S(x)$. We investigate various sampling methods:

167 • The "closest" method deterministically takes the $K$ nearest neighbors.

168 • The "uniform" method samples $K$ neighbors uniformly from the larger neighborhood.

169 • The "biased" method samples proximate neighbors with higher probability than distant neighbors.

170 The loss in equation 3, penalizes local distortion during the alignment, thus preserving the geometry.
171 The choice of a suitable neighborhood encoding to capture local structure is a crucial consideration.
172 We propose to use an approximation of the heat kernel, discussed in Section 4.2.1. Additionally, we
173 report alternative choices and evaluate the differences in performance in Section 5.5.

## 4.2 KERNEL ENCODINGS

### 4.2.1 THE HEAT KERNEL

176 We next present the different choices of kernels to encode the neighborhood structure. Given a set
177 of points, $\{x_i\}_{i=1}^N$, assumed to lie on some low dimensional manifold, $\mathcal{X}$, the diffusion operator
178 (Coifman & Lafon, 2006), denoted by $\mathbf{W}^{\text{Heat}}$, is defined by:

$$
\begin{aligned}
\mathbf{K}^{\text{Heat}}(x_i, x_j) &= e^{-\|x_i - x_j\|_2^2 / 4\epsilon} \\
\mathbf{W}^{\text{Heat}}(x_i, x_j) &= \frac{\mathbf{K}^{\text{Heat}}(x_i, x_j)}{\sum_l \mathbf{K}^{\text{Heat}}(x_i, x_l)}
\end{aligned}
\tag{4}
$$

179 This operator was shown to converge pointwise to the Neumann heat kernel of the underlying data
180 manifold as $\epsilon$ approaches zero and the number of points tends to infinity. Below, we articulate some
181 advantages of using $\mathbf{W}^{\text{Heat}}$ in our formalism:

182 **Intrinsic.** The heat kernel and its approximation are intrinsic, meaning that they are independent of
183 the choice of coordinates. As a result, they are invariant to isometric transformations.

**Informative.** The heat kernel captures essential intrinsic geometric information. For example, the
geodesic distance $g$ between two points $x, y$ on a manifold can be recovered from the heat kernel via
the limit (Varadhan, 1967):

$$
g(x, y) = \lim_{t \to 0} \sqrt{-4t \log h_t(x, y)},
$$

184 where $h_t(x, y)$ denotes the continuous heat kernel, which relates to $\mathbf{W}^{\text{Heat}}$ by $h_t =$
185 $\lim_{\epsilon \to 0, N \to \infty} (\mathbf{W}^{\text{Heat}})^{t/\epsilon}$ (under slightly different normalization) (Coifman & Lafon, 2006).

186 **Multi-Scale.** The locality of the heat kernel is sensitive to the time variable, $t$. In its discrete
187 approximation, $\mathbf{W}^{\text{Heat}}$, the locality is governed by the kernel scale, $\epsilon$, and the sample density of
188 the point cloud. Through these parameters, the heat kernel and its approximation are capable of
189 capturing multi-scale features. Specifically, a smaller $\epsilon$ in equation 4 results in a more local kernel.

### 4.2.2 ALTERNATIVE KERNEL ENCODINGS

191 The majority of our experiments use the diffusion operator to capture local neighborhood geometry,
192 but other choices are possible. For example, the following kernels capture the pairwise $L^2$ distance
193 and related values:

$$
\begin{aligned}
\mathbf{K}^{\text{Linear}}(x_i, x_j) &= \|x_i - x_j\|_2 \quad \forall x_i, x_j \in X \tag{5} \\
\mathbf{K}^{\text{Squared}}(x_i, x_j) &= \|x_i - x_j\|_2^2 \quad \forall x_i, x_j \in X \tag{6} \\
\mathbf{K}^{\text{Inverse}}(x_i, x_j) &= \frac{1}{1 + \|x_i - x_j\|_2^2} \quad \forall x_i, x_j \in X \tag{7}
\end{aligned}
$$

194 We normalize each kernel by the average column values, similarly to the diffusion operator, resulting
195 in the neighborhood encoding $\mathbf{W}_X^Z$, where $Z$ stands for "Linear", "Squared" or "Inverse":

$$
\mathbf{W}^Z(x_i, x_j) = \frac{\mathbf{K}^Z(x_i, x_j)}{\sum_l \mathbf{K}^Z(x_i, x_l)}
\tag{8}
$$

196 In Section 5.5, we empirically demonstrate that the heat kernel yields the best performance, indicat-
197 ing better preservation of local neighborhood information.

## 5 EXPERIMENTS

### 5.1 EXPERIMENTAL DETAILS

We conduct extensive experiments to show the performance of GeRA under limited paired data availability with images and text. In addition, in Section 5.4 we present results with speech and text.

Our default experimental setup is adapted from the setup used in ASIF (Norelli et al., 2022), which serves as a baseline.

**Dataset:** Our training dataset for the image and text experiments is the Conceptual 12M (CC12M) dataset (Changpinyo et al., 2021). This dataset consists of 12 million paired entries of images and their corresponding textual descriptions, spanning a broad spectrum of visual concepts.

**Unpaired points:** To preserve the local geometry of the pretrained unimodal models, we use unpaired points from each modality to compute the geometric regularization in equation 1. For the image and text experiments, we discard the pairing information of $6 \times 10^6$ data points from CC12M and treat them as unpaired points used in the geometric regularization.

**Encoders:** For our first experiments we used the **Vision Transformer (ViT)** (Dosovitskiy et al., 2020) and the **Masked and Permuted Network (MPNet)** (Song et al., 2020). The base model of the ViT has 86 million parameters and the base model of MPNet has 109 million parameters.

**Zero-Shot Accuracy Metric:** We use zero-shot accuracy (Xia et al., 2023) as the metric to assess the quality of our alignment method, measured on ImageNet (Deng et al., 2009). The ImageNet dataset has 1,000 classes, each class is represented by 50 images in the evaluation split. As in Radford et al. (2021), we encode the class names using various prompt templates and average them in the shared embedding space. The images are directly mapped into the common embedding. We calculate the proximity between the image embeddings and the class embedding vectors using cosine similarity. Image classification is determined by computing the nearest class within the embedding space. Clearly, as the alignment method improves, the zero-shot accuracy increases.

**Precision@$k$ Metric:** For evaluation beyond image-text alignment we use the Precision@$k$ metric, applied to the test split of the same dataset used for training. We select 10,000 test pairs resulting in 10,000 classes, such that the samples from one modality form classes, and we attempt their retrieval based on corresponding samples of the other modality, and vice versa. Our findings are reported in terms of precision@1 and precision@5. The test samples remain consistent across all experiments.

### 5.2 BASELINES

We verify the effectiveness of our proposed method by comparing it to established baseline models. First, we examine the Procrustes alignment method, which is designed to learn a rotation matrix that aligns one embedding with another. Then, we assess the performance of our alignment transformation functions when trained solely with the contrastive loss, without including our geometry-preserving regularization method. Lastly, we provide a comparison with the ASIF method, as detailed in the related work section.

### 5.3 IMAGE AND TEXT ALIGNMENT THROUGH GERA

#### 5.3.1 BENCHMARKING ON IMAGENET AND CC12M

We test GeRA with a neighborhood size of $K = 150$, using the heat kernel approximation as the neighborhood encoding scheme, and with the "biased" sampling method. We evaluate our performance compared to the baseline methods on the default configuration as described in section 5.1.

**Results:** Figure 3 shows that GeRA consistently outperforms both Procrustes Alignment and the unregularized alignment based on the contrastive loss. This validates GeRA's design choice of balancing local preservation of geometric structures with global flexibility in the alignment process. GeRA demonstrates a significant improvement of almost 9% over the unregularized alignment. The increase in performance is particularly notable in situations where data availability is highly limited, where accuracy improves from 3% for the contrastive loss trained with 1000 samples to almost 9%.

Figure 3 depicts that GeRA is the best-performing model in the low-data regimes. As the volume of data increases, ASIF slightly outperforms GeRA. However, in Figure 4, GeRA exhibits better results when evaluated on precision@5. In Section 5.7, we further demonstrate the advantage of GeRA over ASIF, in terms of inference time.

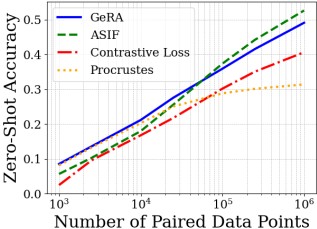 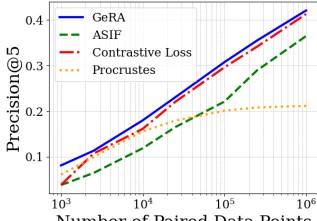 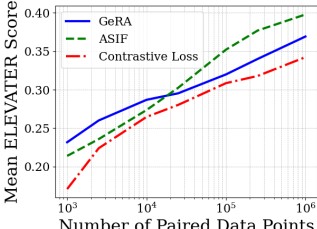

Figure 3: GeRA performance evaluated at zero-shot accruacy on ImageNet against the baselines.

Figure 4: GeRA performance evaluated at precision@5 on in-distribution CC12M data against the baselines.

Figure 5: Comparison using mean scores of GeRA, non-regularized method, and ASIF across 20 ELEVATER datasets.

### 5.3.2 BENCHMARKING ON ELEVATER

To further validate the generality of our method, we expand our experiments beyond ImageNet and CC12M, incorporating multiple vision-language datasets into our evaluation pipeline. We employed the ELEVATER (Li et al., 2022) benchmark, which contains 20 image classification datasets. These datasets cover a broad spectrum of visual concepts, each presenting varying levels of difficulty.

**Results:** Figure 5 demonstrates that GeRA consistently outperforms the non-regularized method on the ELEVATER benchmark. In low-data regimes, the benefits of geometric regularization become especially clear. With 1,000 training pairs, the performance gain exceeds 5%. Even with 1 million training pairs, GeRA delivers an average performance improvement of 2.7%. Considering the diverse visual concepts, it becomes clear that GeRA has superior generalization capabilities.

Compared to ASIF, GeRA demonstrates superior performance in low-data regimes. When trained with 2,500 paired points, GeRA yields a mean score that is almost 3% higher. The advantage of GeRA diminishes as the number of paired points increases; ASIF surpasses GeRA when more paired points are available in training. Specifically, when trained with 1,000,000 paired points, ASIF achieves a mean score 3% higher than GeRA's. However, in this regime, ASIF is more than $100\times$ slower at inference time, as demonstrated in Figure 8.

### 5.4 SPEECH AND TEXT ALIGNMENT THROUGH GERA

To further validate adaptability and performance across diverse modalities, we consider the domain of speech-text alignment. We show that our method's efficacy is not confined to a specific modality and that our hyperparameter choices, optimized for the image-text scenario, are not overfit to that context.

**Encoder:** We use Whisper (Radford et al., 2023) as the speech encoder consisting of 74 million parameters. For text, we again use MPNet (see Section 5.1).

**Dataset:** Our training uses the LibriTTS dataset (Zen et al., 2019). This dataset is an assembly of text-speech pairs, aggregating to 585 hours of read English speech. Each entry corresponds to distinct sentences of speech and their textual counterparts. Entries with significant background noise are filtered out. The dataset includes 205,044 pairs in totals, which is considerably smaller than the text-image alignment dataset of 12 million pairs. In

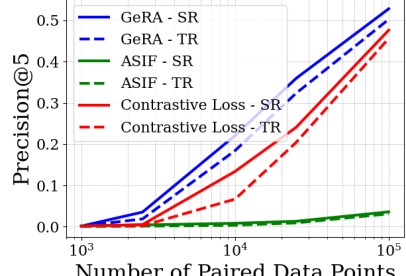

Figure 6: Performance of GeRA compared to ASIF and the pure contrastive learning evaluated at precision@5 for the speech and text alignment, using in-distribution LibriTTS data.

this experiment, we used up to $10^5$ paired points in the contrastive loss, and additional $10^5$ unpaired points in the geometric regularization.

**Results:** Figure 6 shows that GeRA significantly outperforms ASIF on speech–text alignment. The discrepency increases as the number of paired training points increases With 100,000 training pairs, GeRA achieves a precision@5 score for speech retrieval (SR) of over 51% while ASIF's score is below 5%. These results show the generalizing capabilities of GeRA to the speech-text domain, while ASIF struggles with this modality. In addition, GeRA surpasses the model trained solely with the contrastive loss, which attains a precision@5 score of 48% for speech retrieval for 100,000 paired training points.

## 5.5    INFLUENCE OF GEOMETRY PRESERVATION

We analyze the influence of geometric regularization on GeRA via ablation studies measuring the impact of various design choices. These choices include the size of the neighborhood kernel (number of neighbors), the kernel encodings, and the neighborhood sampling method.

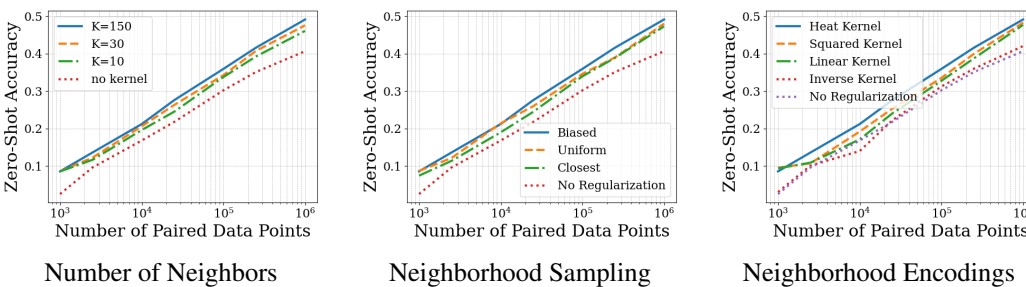

Figure 7: The impact of various design choices in GeRA, including neighborhood kernel size, geometry encoding scheme, and neighborhood sampling method.

### 5.5.1    RESULTS

**Number of Neighbors:** As the kernel matrix size increases, performance improves, but with diminishing returns. Initial increases in neighborhood size yield substantial gains, but this increase plateaus, yielding a trade-off between accuracy and computational cost. We achieve a zero-shot accuracy of over 49% when trained on 1,000,000 paired points using a neighborhood size of 150. Comparing unregularized baseline model to GeRA yields a 9% increase in top-1 accuracy.

**Neighborhood Sampling:** Our experiments show that the sampling method affects accuracy. Biased sampling, using primarily close but also including distant neighbors, proves most effective. The uniform distribution ranks second, including equally close and distant neighbors, whereas the least effective sampling method is the "closest" method, which only includes the nearest neighbors All methods surpass the neighborhood-free baseline.

**Neighborhood Encoding:** Regularization with any of our neighborhood encodings performs better than the contrastive loss alone. Among all, the heat kernel encoding consistently outperforms the other encodings by 2% on average over all training sizes, echoing the theoretical properties inspiring its choice. Overall, our choice of the heat kernel is confirmed to capture geometric information and demonstrates the benefit of geometric regularization in alignment tasks with limited paired data.

## 5.6    INFLUENCE OF PRETRAINED ENCODERS

GeRA is encoder-agnostic and hence not tied to a specific choice of encoders. We initially adopted the configurations that were previously tested with ASIF. Next, we discuss the generality of GeRA across different encoders, demonstrated empirically. See results in Figure 10 in the Appendix.

**Results:** Our method frequently surpasses ASIF by a significant margin with different encoders. This includes CLIP Encoders, the combinations of ViT-RoBERTa, ViT-BERT, and MAE-MPNet. In the setting recommended by ASIF, namely ViT-MPNet or using ViT-SentenceTransformer BERT,

our performance is either on par with or slightly below that of ASIF. Overall, our method offers more consistent and stable results compared to ASIF.

## 5.7 TRAINING AND INFERENCE TIME

In Figure 9, GeRA's training time increases with the neighborhood size used for geometric regularization. Even with the highest number of paired training points (1 million pairs) and the largest kernel size ($K = 150$), however, training GeRA on an NVIDIA GeForce RTX 3090 only takes 20 hours. Figure 8 shows that GeRA has consistent inference times, as the alignment transformation during inference is not affected by neighborhood size or number of training pairs. Conversely, ASIF has significant overhead, with inference times increasing linearly in the number of anchor points. For example, with 1 million training pairs, one ASIF inference takes over 2 hours for the retrieval task, while GeRA completes in under 16 seconds.

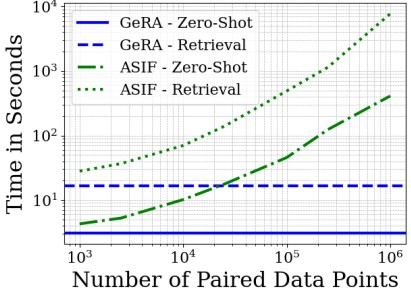
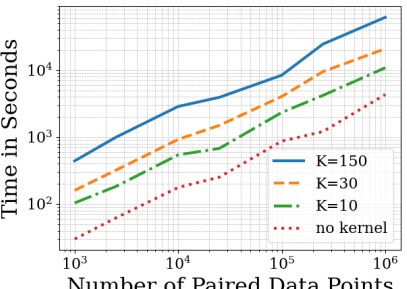

Figure 8: Comparison of inference times between GeRA and ASIF for the Zero-Shot and the Retrieval Evaluation.

Figure 9: Time for training GeRA with different neighborhood sizes using an NVIDIA GeForce RTX 3090.

## 6 DISCUSSION

**Limitations:** Preserving local geometry requires taking neighborhood information into account, which leads to quickly increasing batch sizes, i.e., for batch size $B$ and geometric regularization with $K$ neighbors, the effective batch size becomes $B \cdot K$. This limits the ability to use larger batch sizes, which may slow down the convergence.

Moreover, GeRA depends on powerful pretrained models that define the geometry. In the absence of powerful pretrained models, the regularization's effectiveness diminishes. Our experiments in section 5.6 show that selecting powerful encoders are necessary for both GeRA and ASIF. One potential solution in the absence of powerful pretrained encoders is to collect corrected neighborhood information for our loss term using human annotations or rules defined by a domain expert.

**Future Work:** Our work opens several interesting future work directions. In terms of the attractive capability of GeRA to align domains with limited paired data supervision, there are several other modalities and downstream tasks that could be explored. Examples include aligning protein sequences and biomedical texts, which is needed for protein representation learning. Traditional unimodal approaches, which only focus on protein sequences, often miss functional aspects of proteins. Recent efforts incorporate text data on protein functions as an additional modality, enriching representations (Xu et al., 2023). However, the available datasets are relatively small, featuring only half a million paired data points. As a result, this domain is a key target for future work on GeRA. Additional future work directions for GeRA include exploring learnable parametric geometric kernels (e.g. realized as self-attention blocks or small transformers), simultaneous co-training and multi-task training of both the encoders (on the unimodal data components) and the GeRA alignment module leading to dynamically changing manifolds landscape and potentially requiring exploring into momentum models for increased training stability, exploring multi-scale (coarsening, multi-grid) manifold mapping methods to further enhance the preservation of the more global manifold structure after alignment, and many more.

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

## A  APPENDIX

### A.1  ADDITIONAL DETAILS ON FIGURE 1(B)

The metrics reported in Figure 1(b) are computed based on the models trained in Section 5.3.1 on the CC12M dataset (see results in Figure 3). The model labeled in Figure 1(b) as "No Geom. Regularization" is an alignment model trained with the contrastive loss only, and the modeled labeled as "Geom. Regularization" is an alignment model trained with the GeRA model using $K = 150$ neighbors. Both models were trained using $10^6$ paired points for the contrastive loss, and $6 \times 10^6$ points for the geometric regularization.

In order to further demonstrate the neighborhood distortion effect, we evaluate the image-to-image kNN classification accuracy of ImageNet data using the pretrained image encoder, with and without the alignment layers, trained with the contrastive loss or with GeRA loss as described above. More concretely, we use the validation set of ImageNet, which consists of 50 samples per class. We randomly select 10 samples from each class as labeled training data. We embed this training data and the remaining images using the models. Each image was then assigned to a class based on the majority vote among its $k = 5$ nearest neighbors. We note that different values of $k$ led to similar results. We then compare the performance of GeRA against the unregularized contrastive learning model and the original embedding space generated by ViT before applying our transformation, reported in Table 1. This experiment demonstrates that GeRA preserves the geometry of the image space obtained by the ViT model pre-trained on the image domain, while alignment with vanilla contrastive loss disturbs it.

Table 1: ImageNet kNN accuracy computed in the embedding spaces of ViT, ViT+alignment layers trained with contrastive loss only, and ViT+alignment layers trained with GeRA.

| Method | kNN Classifier Accuracy ($k = 5$) |
|---|---|
| ViT only | 0.76 |
| No Geom. Regularization | 0.67 |
| Geom. Regularization | 0.75 |

### A.2  ADDITIONAL EXPERIMENTAL DETAILS AND HYPERPARAMETERS

#### A.2.1  ADDITIONAL EXPERIMENTAL DETAILS

**Use of Unpaired Points**: To give a bit more detail on the use of unpaired data in our experiments, in the image-text experiments, the dataset we used for training is CC12M which is a paired dataset. However, during training we only consider the pairings for a small number of samples and use the (fraction of) remaining samples as unpaired data to simulate a scenario where there are limited amounts of paired data and many unpaired data points. More concretely, we take $M$ paired samples (used for contrastive loss) and include $N \gg M$ unpaired samples (distinct from the paired points used in the contrastive loss), where the unpaired points for each modality are chosen randomly and independently of the other modality. We leverage the unpaired data in the neighborhoods of each paired datapoint, and construct the kernels in the geometric regularization based on these neighboring unpaired points. Note that for a pair $(x, y)$, the neighborhood for $x$ is in general not the same as the neighborhood for $y$, i.e., the neighbors do not have to be pairs themselves.

**Pre-computing Nearest Neighbors for the Geometric Regularization**: To speed up training time, we pre-compute the neighborhood distributions, $S(x)$, from which $N_k(x)$ is sampled for the geometric regularization in equation 3. For each paired point in each modality, we collect 800 nearest neighbors for constructing $S(x)$. We perform this nearest neighbor search using Faiss (Johnson et al., 2019), which takes approximately $45 - 55$ minutes to compute on an NVIDIA GeForce RTX 3090, for 800 nearest neighbors of $6 \times 10^6$ samples in a 768 dimensional space, takes .

#### A.2.2  HYPERPARAMETERS

**Alpha** ($\alpha$): This parameter balances the geometric regularization term with the contrastive objective, determining the relative importance of each in the loss function (See Equation 1).

Table 2: Summary of the explored hyperparameter spaces and the optimal values discovered for each method. The table outlines the range of values over which each hyperparameter was tuned, denoted in the 'Range' column. Subsequent columns present the hyperparameter values that yielded the best performance for each respective method, GeRA, Contrastive Loss, and ASIF (hyperparameters as stated in the paper (Norelli et al., 2022)), during experimentation. In instances where a hyperparameter is not applicable to a method, the cell is left blank.

| Hyperparameter | Range | GeRA | Contrastive Loss | ASIF |
|---|---|---|---|---|
| Batch Size | 500-4,000 | 2000 | 2000 | – |
| Learning Rate | 1e-5 – 5e-4 | 2e-4 | 2e-4 | – |
| Dropout | 0.0 – 0.5 | 0.3 | 0.3 | – |
| Number of Hidden Layers | 1 – 3 | 1 | 1 | – |
| Hidden Dimension | 768 – 16,000 | 8000 | 8000 | – |
| Output Dimension | 512 – 768 | 768 | 768 | – |
| Number of Neighbors | 5 – 150 | 150 | – | – |
| Alpha | 0.1 – 2.0 | 0.5 | – | – |
| Epsilon ($\sigma$ value) | 0.1 – 3.0 | 0.8 | – | – |
| Temperature | 0.01 – 0.4 | 0.04 | 0.04 | – |
| p (Exponentiation) | 1 – 8 | – | – | 8 |
| k (Sparsification) | 50 – 1600 | – | – | 800 |

**Epsilon** ($\epsilon$): This represents the kernel size, influencing the locality of the kernel matrix. A smaller epsilon value implies that the heat kernel captures more localized features, thereby considering neighbors in closer proximity (See Equation 4). In our experiments with the heat kernel we compute $\epsilon$ by: $\epsilon = \sigma \times \text{mean}\left(\{\|x_i - x_j\|_2^2\}_{i,j}\right)$, i.e., a constant, $\sigma$, multiplied by the mean of the pairwise euclidean distances in the neighborhood. We found that $\sigma = 0.8$ performs best. This kernel normalization adapts to the scale and characteristics of each local neighborhood, and facilitates handling neighborhoods of different sizes and densities.

**Temperature** ($t$): Applied in the output layer, the temperature parameter modulates the sharpness of the distribution. A higher temperature results in a softer probability distribution over classes, whereas a lower temperature makes the distribution more concentrated (See Equation 2).

**Number of Neighbors** ($K$): This parameter specifies the number of neighbors included into the geometric regularization loss (see Equation 3). The larger the amount of neighbors, i.e., the larger the kernel matrix $W$, the better we can capture the local geometry and hence preserve it. Our ablation study in Figure 7 (left) demonstrates that the number of neighbors strongly correlates with the downstream alignment performance. However, the marginal increase in performance seems to diminish with larger neighborhoods, indicating already good performance using relatively small numbers of neighbors.

**Sampling Technique**: We aim to select samples that best represent the local geometry. Hence, selecting only the closest neighbors preserves locality best. However, to obtain better continuity of the embedding space, and increase the amount of information gathered from the neighbors in different epochs, we subsample the neighbors from a larger neighborhood distribution $S(x)$. We examined different ways of sampling the neighbors, including:

- 'Uniform' sampling, where $K$ neighbors are uniformly sampled from the pre-computed neighborhood distribution $S(x)$ (including 800 points in our experiments). This approach includes closer and farther points with equal probabilities.
- 'Closest' sampling, where only the closest $K$ neighbors are chosen from $S(x)$ for each paired point.
- 'Biased' sampling, where $K$ neighbors are sampled from $S(x)$ with higher probabilities given to closer points.

Figure 7 (middle) demonstrates that sampling neighborhood points with a bias towards closest neighbors, i.e., higher probability of sampling closer neighbors while still including some information about further points, performs best overall.

### A.2.3 PERFORMANCE DIFFERENCES OF GERA AND ASIF IN SPEECH-TEXT ALIGNMENT

Figure 6 shows a significant performance gap between GeRA and ASIF in speech-text alignment, in contrary to results for the image-text modality presented in Figure 3, for example. The study by Wang et al. (2023) found that training shared encoders for speech-text data produce more compact and overlapping representations, whereas the embedding spaces of uni-modal encoders yield distinct representations for speech and text.

Taking this observation into account, our results in Section 5.4 highlight the main advantage of our approach over ASIF. When the uni-modal embedding spaces are different, as suggested by Wang et al. (2023) for the speech and text modalities, we hypothesize that ASIF needs a lot more paired samples to properly align the spaces, since extrapolation from a limited number of pairs is likely to be inaccurate. Specifically, the performance obtained by ASIF in Wang et al. (2023), which is better than our reported ASIF performance in this setting, is using encoders that were trained on data that included paired points from the two modalities. In contrast, in our experiments, we use encoders that were trained on purely uni-modal data, and that may be the source of the performance gap.

Unlike ASIF, our method for alignment of the uni-modal models combines the strengths of contrastive alignment with paired data, to match the uni-modal embedding spaces, and preserving the geometry of the respective spaces, thus it is more robust to the differences in uni-modal embedding spaces. For instance, in Figure 6, we see that vanilla contrastive loss performs quite well in aligning (purely) uni-modal models, significantly outperforming ASIF, while our method further improves the performance of the contrastive loss.

### A.3 EVALUATING GERA ON DIFFERENT PRETRAINED ENCODERS

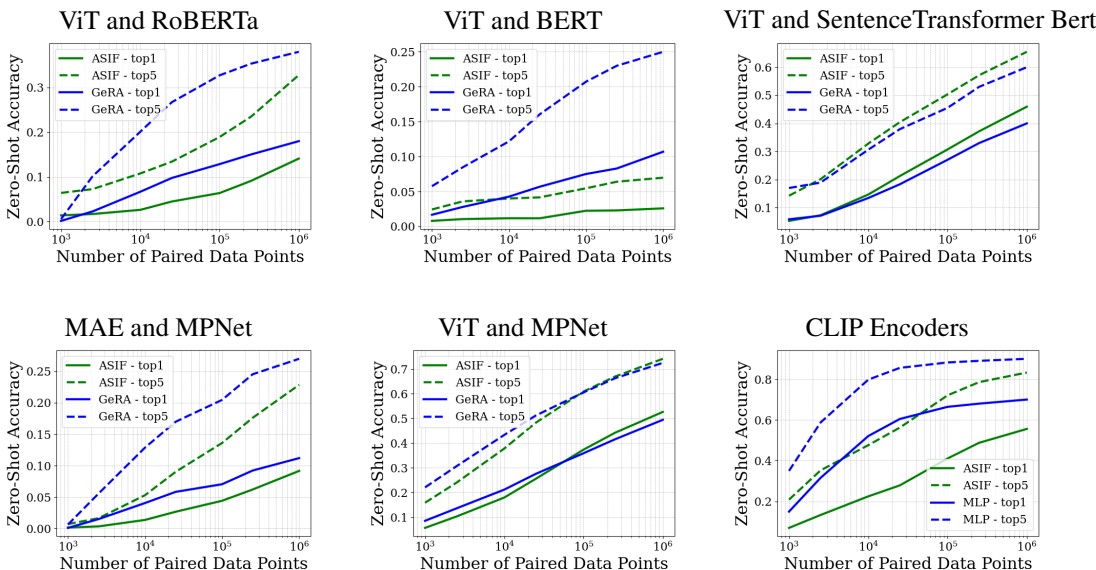

Figure 10: Performance comparison of GeRA and ASIF using various vision and language encoders.

