# OpenReview forum: "GeRA: Label-Efficient Geometrically Regularized Alignment"
_ICLR.cc/2024/Conference — Submitted to ICLR 2024_

### Official Review · Reviewer_gPXV · 2023-10-31

**Soundness:** 3 good
**Presentation:** 4 excellent
**Contribution:** 3 good
**Rating:** 8
**Confidence:** 5

**Summary:**

The manuscript presents a novel semi-supervised method termed as Geometrically Regularized Alignment (GeRA) to effectively align the embedding spaces of pretrained encoders. This approach is characterized by its alignment of two distinct latent spaces into a unified space, leveraging a dual penalty system. The first penalty is a contrastive loss that ensures corresponding points in these spaces are brought close together, while the second, a geometric loss, preserves the inherent local geometry of spaces as learned by the pretrained encoders. A distinguishing feature of this work is the introduction of the geometric loss, which constructs a kernel matrix over neighbors for each sample, aiming to minimize potential distortions when projecting each space in the shared one.

**Strengths:**

- The manuscript stands out for its clarity and coherent flow, providing readers with a well-motivated and well-structured presentation.
- The GeRA method's modality-agnostic nature makes it broadly applicable wherever pretrained models are utilized.
- The introduction of geometric regularization in this context is both original and intuitive, to the best of the reviewer knowledge. The simplicity of the idea that neighboring points in the source spaces should remain neighbors in the aligned space, thus preserving the semantic structures, is a strength of this work.
- The ablation studies presented in the paper are  comprehensive and convincing, enhancing  the presented results.
- The benchmark against ASIF, and subsequently extending this comparison to other pretrained models, is a strong selling point.
- A well-thought hyperparameter search has been executed. Notably, also the competitor ASIF has gone through an hyperparameter search.

**Weaknesses:**

- While the paper reports performance improvements, it lacks clarity on their statistical significance. For instance, when a 3% improvement in performance is highlighted, it becomes crucial to understand the variance arising from different initialization seeds. Reporting standard deviations would have offered a clearer picture of the model's robustness and reliability.
- The manuscript introduces additional hyperparameters, yet it does not provide sufficient insight into their impact on downstream performance. Without guiding intuition, the only approach seems to be extensive hyperparameter tuning, which might be computationally expensive and time-consuming.
- The discussion regarding inference time, particularly in relation to ASIF that performs vector search (i.e. retrieval) using cosine similarity, may be misleading. Existing libraries, such as faiss [1], offer methodologies for more efficient vector searches (e.g., approximate or hierarchical techniques). As such, the paper's emphasis on superior performance compared to a naive ASIF implementation could potentially be misleading to the reader.
- There is no publicly available code to reproduce the work.

Overall, I think this paper is good enough to be accepted. However, the work would benefit from further validation, specifically regarding the statistical significance of the improvements reported. *The reviewer is ready to further raise the score if the authors present additional evidence regarding the statistical significance.*

---

[1] Johnson, Jeff, Matthijs Douze, and Hervé Jégou. 2017. “Billion-Scale Similarity Search with GPUs.”, IEEE Transactions on Big Data.

**Questions:**

- The manuscript does not discuss the interplay between the density of the latent space and the preservation of its geometry. In particular for spaces where the points density changes in different regions, it may be problematic to use the same neighbors selection strategy for the whole space
 - The paper presents notably low performance figures for ASIF in the domain of speech-text alignment. How do these figures relate to the findings of [2] when using ASIF? An explanation or a comparative insight would help in contextualizing the reported results better and understanding potential disparities or improvements.

---

[2] Gary Wang, Kyle Kastner, Ankur Bapna, Zhehuai Chen, Andrew Rosenberg, Bhuvana Ramabhadran, and Yu Zhang. Understanding shared speech-text representations. In ICASSP 2023-2023  IEEE International Conference on Acoustics, Speech and Signal Processing (ICASSP)

---

> ### Author Response · Authors · 2023-11-17
> **Responses to Reviewer Questions**
>
> We thank the reviewer for their questions and comments. Please see our responses below.
>
> > While the paper reports performance improvements, it lacks clarity on their statistical significance. For instance, when a 3% improvement in performance is highlighted, it becomes crucial to understand the variance arising from different initialization seeds. Reporting standard deviations would have offered a clearer picture of the model's robustness and reliability.
>
> We conducted 5 runs with 100K paired points for both GeRA and unregularized contrastive learning, of the experiment presented in Figure 3. We report the mean and standard deviation below.
>
> GeRA:
> MEAN:  0.3507
> STD: 0.0063
>
> Contrastive Loss:
> MEAN: 0.3105
> STD: 0.0048
>
> ASIF is not probabilistic and returns constant results.
>
> This indicates that the performance of GeRA is insensitive to initialization and the improvement over vanilla contrastive loss is significant. Obtaining standard deviations for all experiments takes time and is not feasible within the discussion period, but we will add standard deviations to our figures in the final draft.
>
> > The manuscript introduces additional hyperparameters, yet it does not provide sufficient insight into their impact on downstream performance. Without guiding intuition, the only approach seems to be extensive hyperparameter tuning, which might be computationally expensive and time-consuming.
>
> The main additional hyperparameters in comparison to vanilla contrastive loss that may require some tuning are $\alpha$ (balancing the contrastive loss term and the geometric loss term) and kernel hyperparameter $\epsilon$. The choice of $\alpha$ signifies the degree to which we focus on preserving local geometry at the expense of potentially reducing the alignment quality. The kernel scale $\epsilon$ defines the notion of neighborhood size in the kernel. We set it to equal to the mean of the pairwise distances in the neighborhood, multiplied by a constant, which is the tuned hyper-parameter. Setting a lower constant puts more weight on similarities of closer neighbors in the kernel construction, decreasing the effective neighborhood size in the kernel (see equation 4) and focusing on more localized features.
>
> All other introduced hyperparameters have clear trends in terms of their effect on performance as demonstrated by our ablation studies, presented in Figure 7. We summarize the results below:
>
> * *Number of Neighbors*: Most significant is the size of the neighborhood. The intuition here is the more samples we have in our neighborhood, i.e., the larger the kernel matrix $W$, the better we can capture the local geometry and hence preserve it. Figure 7 (left) clearly shows that the number of neighbors strongly correlates with the downstream alignment performance. However, the marginal increase in performance seems to diminish with larger neighborhoods, indicating already good performance using relatively small numbers of neighbors.
>
> * *Sampling Technique*: We examined different ways of sampling the neighbors and found that the sampling technique has an effect on performance. We aim to select samples that best represent the local geometry. Hence, selecting only the closest neighbors preserves locality best. However, to obtain better continuity of the embedding space, and increase the amount of information gathered from the neighbors in different epochs, we subsample the neighbors from a larger neighborhood. Figure 7 (middle) demonstrates that sampling neighborhood points with a bias towards closest neighbors, i.e., higher probability of sampling closer neighbors while still including some information about further points, performs best overall.
>
> * *Kernel Type*: Figure 7 (right) shows that the choice of the kernel type, i.e., the neighborhood encoding method affects performance as well. This plot demonstrates that the heat kernel is the best performing kernel, indicating its superiority in capturing the local geometry. This is also underlined by the theoretical properties of the heat kernel that can capture the geometry of the manifold and hence meets our expectations.
>
> We added these clarifications and details to Appendix A.2.2 in the paper as well.

---

> ### Author Response · Authors · 2023-11-17
> **Responses to Reviewer Questions - Continue**
>
> > The discussion regarding inference time, particularly in relation to ASIF that performs vector search (i.e. retrieval) using cosine similarity, may be misleading. Existing libraries, such as faiss [1], offer methodologies for more efficient vector searches (e.g., approximate or hierarchical techniques). As such, the paper's emphasis on superior performance compared to a naive ASIF implementation could potentially be misleading to the reader
>
> To address this question, we examine the performance of ASIF with Faiss for neighborhood computation in terms of accuracy and computation time, leveraging Faiss's efficiency for neighbor and similarity computation. For comparability, we mirrored the approximation conditions that we already used when precomputing the nearest neighbors for the GeRA method. This involved quantizing vectors using 64 bits and employing 50 clusters with ‘nprobe=5’. The following table shows the resulting zero-shot accuracies on the ImageNet data for different numbers of paired points ($M$) (an experiment similar to the one depicted in Figure 3), with and without faiss:
> | Method | $M=10^3$ | $M=2.5\times 10^3$ | $M=10^4$ | $M=2.5\times 10^4$ | $M=10^5$ | $M=2.5\times 10^5$ | $M=10^6$ |
> |-----------------|------------|------------|------------|------------|------------|------------|------------|
> | ASIF | $0.06$ | $0.1$ | $0.18$ | $0.26$ | $0.37$ | $0.44$ | $0.53$ |
> | ASIF+faiss | $0.04$ | $0.06$ | $0.11$ | $0.17$ | $0.22$ | $0.22$ | $0.28$ |
>
> This table depicts that the downstream performance of ASIF drops significantly when using faiss.
>
> In addition, in terms of computation times, using faiss for computing sparse vector representations in ASIF did not improve the overall run times for the number of points considered (up to $10^6$). This is mainly due to the computation of cosine similarities between the representations of the different modalities for each new input point, which requires computing intersection of indices that cannot be vectorized. This approach will be beneficial when there are significantly more paired points.
>
> While other ways of implementing ASIF more efficiently could be explored further, considering the substantially decreased zero-shot performance presented in the table, the run-time discussion of ASIF in the paper is fair.
>
> > There is no publicly available code to reproduce the work
>
> We will publish the code on acceptance
>
> > The manuscript does not discuss the interplay between the density of the latent space and the preservation of its geometry. In particular for spaces where the points density changes in different regions, it may be problematic to use the same neighbors selection strategy for the whole space
>
> Thank you for pointing this out. We added more details about how we compute the kernel scale $\epsilon$ to Appendix A.2.2, which addresses this point. In order to allow better adaptability of the kernel to neighborhoods of different sizes and densities, we compute the kernel scale $\epsilon$ as follows: $\epsilon = \sigma \times\mathrm{mean}(\mathrm{pairwise\  euclidean\  distances})$, where $\sigma$ is the tuned hyperparameter (constant).
>
> Hence, by including the average distances within each neighborhood in the kernel normalization, the kernel adapts to the local scales and characteristics of these neighborhoods.
>
> > The paper presents notably low performance figures for ASIF in the domain of speech-text alignment. How do these figures relate to the findings of [2] when using ASIF? An explanation or a comparative insight would help in contextualizing the reported results better and understanding potential disparities or improvements
>
> Thank you for pointing out this relevant paper. Our results and the results presented in [2] using ASIF highlight the main advantage of our approach over ASIF. When the uni-modal embedding spaces are different, as suggested by [2] for the speech and text modalities, we hypothesize that ASIF needs a lot more paired samples to properly align the spaces, since extrapolation from a limited number of pairs is likely to be inaccurate. Moreover, the performance obtained by ASIF in [2] is using encoders that were trained on data that included paired points from the two modalities. In contrast, in our experiments, we use encoders that were trained on purely uni-modal data, and that may be the source of the performance gap. Our method for alignment of the uni-modal models combines the strengths of contrastive alignment with paired data, to match the uni-modal embedding spaces, and preserving the geometry of the respective spaces, thus it is more robust to the differences in uni-modal embedding spaces. For instance, in Figure 6, we see that vanilla contrastive loss performs quite well in aligning uni-modal models, significantly outperforming ASIF (unlike the text-image experiment in Figure 3), while our method further improves the performance of contrastive loss.
>
> We added this discussion and citation of [2] to Appendix A.2.3 in the paper

---

> > ### Comment · Reviewer_gPXV · 2023-11-20
> >
> > Thank you for the detailed rebuttal provided in response to the comments and questions regarding the manuscript.  The responses have not only clarified all the concerns but also enhanced the overall work.
> >
> > Thus, I am increasing my score and firmly reccommending acceptance.

---

### Official Review · Reviewer_G2FF · 2023-11-01

**Soundness:** 3 good
**Presentation:** 3 good
**Contribution:** 3 good
**Rating:** 5
**Confidence:** 4

**Summary:**

This paper aims to improve the training of multi-modal models from pretrained unimodal models. The paper proposes the method GeRA that adds a regularization loss to the multi-modal contrastive loss such that the local geometry of the original embedding spaces of each modality is preserved (Eq. 3: geometric regularization). The regularization term is based on a kernel function of the locality around each point that encourages the nearest neighbors to stay in their relative positions. The experiments section evaluates the effectiveness of the geometric regularization compared with various baselines on the CC12m and concludes that the method is effective in a low data regime.

**Strengths:**

- Results in Figure 3 and Figure 6 show that the proposed method is better than training with contrastive loss and two other baselines when training on fewer than 10^5 paired data points on both image-text and speech-text alignment tasks.
- GeRA has been shown to be effective for two alignment learning tasks, image-text and speech-text alignment, in the low-data regime.

**Weaknesses:**

- One of the major motivations in the introduction for the method is to use unpaired data for training. However, I cannot find any experiment in section 5 that trains on a mixture of unpaired and paired data where paired data is small. If so, please name the dataset used in section 5. Is unpaired data referring to the data used for pretraining the models? If no experiments are done that use unpaired data during the alignment, at least the following sentence in the abstract should be corrected: “Our method leverages the manifold geometry of unpaired (unlabeled) data to improve alignment performance.” and the following sentence in Section 3: “...we propose to leverage unlabeled (unpaired) points from each modality to preserve the rich geometric structure of their original embedding spaces.”
- The effectiveness of the method is limited to the low-data regime with paired data fewer than 10^5 samples and the performance is significantly lower than a model trained with just one order of magnitude more paired samples. So at least for image-text and speech-text modalities where available paired data is significantly more than 10^5, it is not clear how the proposed method can be helpful. In other words, when would it be useful to use GeRA instead of ASIF or the standard contrastive loss? Would it be for training other multi-modal models where the paired data is few? If so, do authors believe that the two examples of image-text and speech-text can be extrapolated to other multi-modal models?

**Questions:**

- Figure 1: This figure is interesting and shows that the nearest neighbors remain relatively the same. Is there a standard evaluation metric that is improved because the geometry of modalities remains almost the same? Most of section 5 considers evaluation metrics specific to multi-modal models. Can we also evaluate these models for unimodal metrics such as unimodal retrieval such as image-image text-text retrieval?
- Is Figure 1 related to any model trained and evaluated in Section 5? Can we confirm that the model trained with GeRA is also a strong model according to zero-shot metrics?
- Eq. 3: What is the dimensionality of W? Is it MxM or N_k x N_k? Is the loss meaningful even if the nearest neighbors in the original and the new space are different?

---

> ### Author Response · Authors · 2023-11-17
> **Responses to Reviewer Questions**
>
> We thank the reviewer for their questions and comments. Please see our responses below.
>
> > One of the major motivations in the introduction for the method is to use unpaired data for training. However, I cannot find any experiment in section 5 that trains on a mixture of unpaired and paired data where paired data is small. If so, please name the dataset used in section 5. Is unpaired data referring to the data used for pretraining the models?
>
> Thank you for pointing out this source of confusion. We added a clarification in Section 5.1 in the paper (paragraph Unpaired points) and in Section 5.4 (paragraph Dataset) that in all experiments we make use of both paired points, in the contrastive loss, and unpaired points, in the geometric loss when training the alignment neural networks:
>
> Section 5.1: “To preserve the local geometry of the pretrained unimodal models, we use unpaired points from each modality to compute the geometric regularization in equation 1. For the image and text experiments, we discard the pairing information of $6\times 10^6$ data points from CC12M and treat them as unpaired points used in the geometric regularization.”
>
> Section 5.4: “In this experiment, we used up to $10^5$ paired points in the contrastive loss, and additional $10^5$ unpaired points in the geometric regularization.”
>
> In addition, we add the following clarification in Appendix A.2.1: “To give a bit more detail on the use of unpaired data in our experiments, in the image-text experiments, the dataset we used for training is CC12M which is a paired dataset. However, during training we only consider the pairings for a small number of samples and use the (fraction of) remaining samples as unpaired data to simulate a scenario where there are limited amounts of paired data and many unpaired data points. More concretely, we take $M$ paired samples (used for contrastive loss) and include $N \gg M$ unpaired samples (distinct from the paired points used in the contrastive loss), where the unpaired points for each modality are chosen randomly and independently of the other modality. We leverage the unpaired data in the neighborhoods of each paired datapoint, and construct the kernels in the geometric regularization based on these neighboring unpaired points. Note that for a pair $(x,y)$, the neighborhood for $x$ is in general not the same as the neighborhood for $y$, i.e., the neighbors do not have to be pairs themselves.”
>
> > The effectiveness of the method is limited to the low-data regime with paired data fewer than 10^5 samples and the performance is significantly lower than a model trained with just one order of magnitude more paired samples. So at least for image-text and speech-text modalities where available paired data is significantly more than 10^5, it is not clear how the proposed method can be helpful. In other words, when would it be useful to use GeRA instead of ASIF or the standard contrastive loss? Would it be for training other multi-modal models where the paired data is few? If so, do authors believe that the two examples of image-text and speech-text can be extrapolated to other multi-modal models?
>
> The method presented in this work is model-agnostic and can therefore be applied to any combination of modalities, given good pre-trained unimodal models. GeRA is especially useful in modality pairs for which there are limited amounts of paired data points, as demonstrated by Figures 3 through 6, for relatively few paired data points.
>
> We demonstrate our approach on the image-text and speech-text modality pairs due to their clear ground truth, but it can be applied to any modality pair. Hence, as pointed out correctly by the reviewer GeRA will be especially useful for modalities where the amount of paired data is low.
> Comparing to ASIF, there are two limiting factors:
>
> 1. In the original paper, ASIF's evaluation was limited to the ViT image encoder - a very specific image encoder - that was partially trained on ImageNet images during pretraining. In Fig 10 we use different pretrained encoders. These results show that, compared to GeRA, ASIF struggles much more with other choices of encoders as well as when it comes to retrieving CC12M images and captions (See Figure 4).
>
> 2. ASIF has much higher inference time than GeRA.

---

> > ### Author Response · Authors · 2023-11-17
> > **Responses to Reviewer Questions - Continue**
> >
> > > Is Figure 1 related to any model trained and evaluated in Section 5? Can we confirm that the model trained with GeRA is also a strong model according to zero-shot metrics?
> >
> > The metrics reported in Figure 1(b) are computed based on the same models used for comparing zero-shot performance in Figure 3. Specifically, in Figure 1 we are using the unregularized model with only the contrastive loss (labeled ’’No Geom. Regularization’’) and the GeRA model with $K=150$ neighbors (labeled ``Geom. Regularization’’). Both models were trained with $10^6$ paired points. Figure 3 confirms that GeRA performs well according to the zero-shot metric as well. We added more detail on the creation of Figure 1(b) in Appendix A.1 of the paper.
> >
> > > Figure 1: This figure is interesting and shows that the nearest neighbors remain relatively the same. Is there a standard evaluation metric that is improved because the geometry of modalities remains almost the same? Most of section 5 considers evaluation metrics specific to multi-modal models. Can we also evaluate these models for unimodal metrics such as unimodal retrieval such as image-image text-text retrieval?
> >
> > To address this question, we conducted a new experiment using an alignment model trained with GeRA and the unregularized model trained only with the contrastive loss. Both models were trained using $10^6$ paired points. Note that these are the same models used to create Figure 3.
> > The unimodal metric that we compute is the image-to-image kNN accuracy on ImageNet. More concretely, we took the validation set of ImageNet which consists of 50 samples per class. We randomly selected 10 samples from each class as labeled training data. We embed the training images and the remaining images using our models. Each image was then assigned to a class based on the majority vote among its nearest 5 neighbors (results with 7 and 9 neighbors were similar). We then compared the performance of GeRA against the unregularized contrastive learning model and the original embedding space generated by ViT before applying our transformation.
> > | Method | $k=5$ |
> > |-------------------------|---------|
> > | ViT only | $0.76$ |
> > | No Geom. Regularization | $0.67$ |
> > |  Geom. Regularization | $0.75$ |
> >
> > This experiment demonstrates that GeRA preserves the geometry of the image space obtained by the ViT model pre-trained on the image domain, while alignment with vanilla contrastive loss disturbs it.
> > We added the description and results of this experiment to Appendix A.1 in the paper.
> >
> > > Eq. 3: What is the dimensionality of W? Is it MxM or N_k x N_k? Is the loss meaningful even if the nearest neighbors in the original and the new space are different?
> >
> > The dimensionality of $W$ is $N_k \times N_k$, as it computes the local neighborhood structure for each sample involved in the contrastive loss based on its k-nearest neighbors, which include unpaired data.
> > The aim of the geometric regularization in the loss is to enforce that the nearest neighbors in the original space and in the new space remain similar (as depicted by Figure 1). For this purpose, we compute the neighbors based on the original space only, and use the *same set of neighbors* in the geometric loss for both the original and the new space, thus encouraging an alignment that preserves the local neighborhood structure.

---

> > > ### Comment · Reviewer_G2FF · 2023-11-22
> > > **I thank the authors. Some of my concerns are resolved, however, three concerns need more discussions.**
> > >
> > > I thank the authors for their response. Some of my concerns are resolved, however, three concerns under W.2 Q.1 and Q.3 would need more discussions.
> > >
> > > **W.1** I cannot find any experiment … that trains on a mixture of unpaired and paired data.
> > > The added text in Section 5.1 and Section 5.4 clarifies it.
> > >
> > > **W.2** The effectiveness of the method is limited to the low-data regime. When would it be useful to use GeRA …?
> > > I thank the authors for confirming my understanding. I would count the limited scale of experiments (diversity of datasets and modalities) as a weakness but I don’t count it as a major weakness.
> > >
> > > **Q.1** Is there a standard evaluation metric that is improved? Can we evaluate these models for unimodal metrics such as unimodal retrieval?
> > > New results in the rebuttal partially answers this question. I recommend performing the standard image-image retrieval and text-text retrieval evaluations and report recall@1 separately for all models.
> > >
> > > **Q.2** Can we confirm that the model trained with GeRA is also a strong model according to zero-shot metrics?
> > > The added text in A.1 and the rebuttal comment clarifies it.
> > >
> > > **Q.3**  What is the dimensionality of W? Is the loss meaningful even if the nearest neighbors in the original and the new space are different?
> > > The answer says the N_k refers to a set of k neighbors in the original space.  In that case, the geometric loss depends on the embeddings of data points that don’t exist in a mini-batch if we randomly sample training data. Is there a lookup table that stores all the embeddings of NNs for all points? Does one need to include all the original NNs for all points in a mini-batch? This leads me to a new question and potential concern: **What is the cost of computing the geometric regularization?**

---

> ### Author Response · Authors · 2023-11-22
> **Response to Reviewer's Comment**
>
> We thank the reviewer for the comment. Please see our responses below.
>
> > The effectiveness of the method is limited to the low-data regime.
>
> While mostly motivated by low-data scenarios, GeRA can also take advantage of more data. This is indicated by Figure 3-Figure 7, in which GeRA consistently outperforms the vanilla CL loss, which was used to train models like CLIP, when increasing the number of paired points. However, studying GeRA in scenarios with very large paired datasets is out of the scope of this paper, since we focus on facilitating better alignment specifically in the low data regime.
>
> > Is there a standard evaluation metric that is improved?
>
> In uni-modal self-supervised literature, using uni-modal model embeddings for classification is standard. Our ImageNet experiment, conducted in response to the reviewer's query, demonstrates that GeRA enhances performance over unregularized contrastive learning.
>
> > Can we evaluate these models for unimodal metrics such as unimodal retrieval? New results in the rebuttal partially answers this question. I recommend performing the standard image-image retrieval and text-text retrieval evaluations and report recall@1 separately for all models.
>
> Our main focus is on multi-modal alignment, and we believe multi-modal metrics should be the primary concern to evaluate the performance of alignment transformation functions. While we acknowledge that uni-modal metrics are useful for analyzing neighborhood consistency, our experiments in Figure 1 and the image-to-image kNN accuracy on ImageNet demonstrate that GeRA's alignment transformation functions achieve superior performance in uni-modal metrics for in-distribution data and in zero-shot settings. Furthermore, the results of this experiment are indicative of uni-modal retrieval and recall@1 metrics, due to the use of k-NN (examined with different 'k' values), which relies on the proximity of the images in the embedded space. We will additionally investigate the uni-modal classification performance with more models and linear probing in the final draft.
>
> > Is there a lookup table that stores all the embeddings of NNs for all points? Does one need to include all the original NNs for all points in a mini-batch?
>
> We use the Faiss library [1] to precompute the nearest neighbors for each sample. With approximation methods, this process takes approximately 50 minutes for 6 million embedding points. Incorporating these neighbors into the minibatch for geometric computation does introduce additional computational overhead. However, as shown in Figure 9, which details the training time of our geometrically regularized alignment, training is completed within a reasonable time. It takes 20 hours even with 150 neighbors, using a single NVIDIA GeForce RTX 3090 GPU.
>
> [1] Johnson, Jeff, Matthijs Douze, and Hervé Jégou. 2017. “Billion-Scale Similarity Search with GPUs.”, IEEE Transactions on Big Data.
>
> > What is the cost of computing the geometric regularization?
>
> The computational cost of geometric regularization scales asymptotically with the square of the neighborhood points count in the geometrical loss computation. However, this cost is relatively minor compared to the computational graph's construction during backpropagation which has a larger computational constant. In our experiments, the overhead increased approximately linearly with the number of neighbors per batch. Figure 9 in our paper illustrates the training time for GeRA across various neighborhood sizes. We will also include a line for training time without geometric regularization as a reference.

---

> > ### Author Response · Authors · 2023-11-22
> > **Including Training Time without Geometric Regularization**
> >
> > We've revised Figure 9 to now show the training time for alignment transformation without geometric regularization. While a larger neighborhood size does extend the training time, the difference between pure contrastive learning and GeRA, particularly with a smaller kernel size, remains modest. Notably, Figure 7 shows that marginal increase in performance seems to diminish with larger neighborhoods, indicating already good performance using relatively small numbers of neighbors.

---

### Official Review · Reviewer_cT78 · 2023-11-01

**Soundness:** 2 fair
**Presentation:** 3 good
**Contribution:** 2 fair
**Rating:** 5
**Confidence:** 2

**Summary:**

The paper presents a method for addressing the challenge of multi-modal alignment with a focus on preserving local geometric structure and efficiently utilizing unlabeled data. The paper makes several contributions to the field, including geometry-preserving alignment, label efficiency, and modality-agnostic formulation. The authors demonstrated the effectiveness of the proposed GeRA method in various settings

**Strengths:**

The proposed method stands out by focusing on preserving local geometric structures, which are critical for retaining the rich semantic information within the manifold structure.

The method can capture additional information from pretrained unimodal encoders, making it highly valuable in scenarios where paired data is limited.

does not rely on domain-specific knowledge or augmentation and can be applied across various encoders and data modalities, as long as pretrained models are available.

**Weaknesses:**

The author proposed the kernel based encoding methods for capturing the local geometric information of each sample. There are several existing works proposed in a while for capturing the local geometric in RKHS in semi-supervised settings, through either constructing neighbor data dependent norms or leveraging the Laplacian graphs in manifold regularization, list a few below:

V. Sindhwani, et al.   Beyond the point cloud: from transductive to semi-supervised learning

X. Zhu, et al.   Semi-supervised learning using gaussian fields and harmonic functions

From this point of view, the employment of the heat kernel seems to be the main contribution of this work, thus slightly weakening the novelty.
As the author mentioned, there is a clear limitation related to the batch size and computational cost of this method, have the author conducted any analysis based on what could be the trade off due to this limitation? What could be the scenario in which this method may not work well due to this?

**Questions:**

As listed in weakness

---

> ### Author Response · Authors · 2023-11-17
> **Responses to Reviewer Questions**
>
> We thank the reviewer for their questions and comments. Please see our responses below.
>
> > The author proposed the kernel based encoding methods for capturing the local geometric information of each sample. There are several existing works proposed in a while for capturing the local geometric in RKHS in semi-supervised settings, through either constructing neighbor data dependent norms or leveraging the Laplacian graphs in manifold regularization, list a few below:
> V. Sindhwani, et al. Beyond the point cloud: from transductive to semi-supervised learning
> X. Zhu, et al. Semi-supervised learning using gaussian fields and harmonic functions
> From this point of view, the employment of the heat kernel seems to be the main contribution of this work, thus slightly weakening the novelty.
>
> The papers mentioned by the reviewer are highly relevant and we thank the reviewer for pointing them out. We added the following to the `related work’ section in the paper (starting in line 108):
>
> “In the context of semi-supervised learning, Sindhwani, et al., Zhu, et al., propose frameworks for integrating geometry learned from both labeled and unlabeled data into classification algorithms based on the graph Laplacian (Sindhwani, et al.), and based on a Gaussian random field model (Zhu, et al.). These works, however, focus on a uni-modal supervised learning setting and do not address semi-supervised alignment of data from multiple modalities.”
>
> These papers are related in the sense that they address the incorporation of unlabeled points into kernel methods to facilitate semi-supervised learning, but both focus on learning purely within a single modality in a supervised learning setting. In contrast, we focus on the objective of multi-modal alignment with limited paired data, and hence the main novelty of our paper is not the geometric regularization term itself, but rather the whole framework. We achieve our main objective of aligning multi-modal data by combining two concepts that have not been studied together so far: (1) alignment of uni-modal pre-trained models and (2) a geometric regularization term aimed at preserving local neighborhood structures of the uni-modal models.
>
> The use of geometric information in learning tasks is indeed a highly studied topic, yet the effective incorporation and design of geometric regularizers for different learning problems, especially involving deep learning, is non-trivial.
>
> > As the author mentioned, there is a clear limitation related to the batch size and computational cost of this method, have the author conducted any analysis based on what could be the trade off due to this limitation? What could be the scenario in which this method may not work well due to this?
>
> The computational limitations stem from the number of neighbors used in the kernel construction. We demonstrated that using more neighbors improves downstream performance (see Figure 7). However, increasing the number of neighbors limits the number of paired points used for the contrastive loss part in each batch. Therefore, the main trade-off is between performance gained by increasing the batch size of paired points in the contrastive loss, and the performance gained by using neighborhood geometry as a regularizer. Figure 7 demonstrates that there is a large increase in performance when adding the regularization term, even for small neighborhoods, and then a small marginal increase for larger neighborhoods. Therefore, using smaller neighborhoods for kernel construction reduces computational overhead, while preserving most of the benefits obtained by using the geometric regularization.

---

### Official Review · Reviewer_GTA6 · 2023-11-01

**Soundness:** 3 good
**Presentation:** 3 good
**Contribution:** 2 fair
**Rating:** 5
**Confidence:** 4

**Summary:**

This paper proposes a new semi-supervised method for cross-modality alignment, named Geometrically Regularized Alignment (GeRA). Compared with regular aligning loss, GeRA includes Geometric Regularization, which force to preserve the neighborhood structure of nearby unpaired points.

**Strengths:**

- The whole paper is well written and easy to follow.
- To effeciently align embedding spaces of unimodal encoders by preserving the locality of unparied points is convincing.
- The figures and charts are well-presented. Both Fig1 and Fig2 illustrates GeRA clearly.

**Weaknesses:**

- This paper has limited novelty. Adding a geometrically regularization term is too conventional in manifold learning.
- The motivation of this paper needs further discussion. There are millions or even billions of paired  speech-text and image-text  data, why do we need a label-efficient semi-supervised method?

**Questions:**

- Please explain my questions mentioned in Weakness.
- How long does it take to get the nearest neighbor information?

---

> ### Author Response · Authors · 2023-11-17
> **Responses to Reviewer Questions**
>
> We thank the reviewer for their questions and comments. Please see our responses below.
>
> > This paper has limited novelty. Adding a geometrically regularization term is too conventional in manifold learning.
>
> Our objective in this paper is to align multiple modalities given limited amounts of paired data. From this perspective, the main novelty of our paper is not the geometric regularization term itself, but rather the entire framework.
> Our work combines two concepts that have not previously been studied together: preserving uni-modal geometry using a geometric loss term and multi-modal alignment based on uni-modal models. When the amount of paired data is limited, a multi-modal model cannot be sufficiently trained from scratch. In addition, we demonstrate that aligning embedding spaces of pre-trained unimodal models without taking into account the local geometry degrades performance and distorts neighborhood structures. We thus propose a new semi-supervised framework for aligning pre-trained unimodal models that preserves the local neighborhood structures. This resultes in improved performance in multi-modal zero-shot and retrieval tasks.
>
> Secondarily, geometric regularization is indeed a pillar of machine learning theory and practice, and the design of effective geometric regularizers for different learning problems is far from trivial. The use of this time-tested and effective strategy for a new problem in machine learning should not be grounds for rejection, especially since our particular regularization technique is new.
>
> > The motivation of this paper needs further discussion. There are millions or even billions of paired speech-text and image-text data, why do we need a label-efficient semi-supervised method?
>
> Our approach is model-agnostic and can therefore be applied to any combination of modalities, given good pre-trained unimodal models. GeRA is indeed especially useful in modality pairs for which there are limited amounts of paired data points, as demonstrated by Figures 3 through 6, for relatively few paired data points. In this paper, we demonstrate our approach on relatively small datasets of the image-text and speech-text modality pairs due to their availability and clear ground truth, however, it can be applied to any modality pair.
>
> > How long does it take to get the nearest neighbor information?
>
> Using Faiss [*] for computing 800 nearest neighbors for 6 millions vectors in a 768 dimensional space it takes 45 to 55 minutes on a NVIDIA GeForce RTX 3090. We added these details to Appendix A.2.1 in the paper as well.
>
> [*] Johnson, Jeff, Matthijs Douze, and Hervé Jégou. 2017. “Billion-Scale Similarity Search with GPUs.”, IEEE Transactions on Big Data.

---

> > ### Comment · Reviewer_GTA6 · 2023-11-21
> >
> > Thank you to the authors for their detailed explanation. However, I still have a major concern.
> >
> > The authors stated, "Our objective in this paper is to align multiple modalities with limited paired data. The main novelty of our work lies not in the geometric regularization term itself, but in the entire framework." However, the evaluation of this framework was conducted using speech-text and image-text data, which typically have an abundance of paired data. To effectively assess the framework's utility, it would be crucial to test it in scenarios where paired data across multiple modalities are genuinely scarce.
> >
> > Upon a thorough review of the entire rebuttal and the comments from other reviewers, I noticed that reviewer G2FF also identified this as an issue. Consequently, I have decided to keep my original rating unchanged."

---

> > > ### Author Response · Authors · 2023-11-22
> > > **Response to Official Comment by Reviewer GTA6**
> > >
> > > We thank the reviewer for the comment.
> > >
> > > It is a standard practice in ML literature for semi-supervised/self-supervised methods to re-use standard datasets such as ImageNet, despite labels on this dataset being readily available. For example, SimCLR [*], one of the pioneering self-supervised learning methods performs extensive evaluation on ImageNet, despite availability of ImageNet labels and the superior performance of standard supervised learning. We also note that our main baseline, ASIF, only considers image-text modality pair, despite existing powerful pre-trained image-text models being available. In our work, we additionally perform experiments on the speech-text modality.
> > >
> > > We agree that experiments on domains where paired data is scarce would be interesting, however such domains (e.g., protein sequences mentioned in the conclusion) typically require subject expertise and a dedicated paper to conduct a meaningful study of a new AI method. We are currently exploring such opportunities and reaching out to subject experts, however this is beyond the scope of this paper.
> > >
> > > [*] Chen, T., Kornblith, S., Norouzi, M., & Hinton, G. (2020). A Simple Framework for Contrastive Learning of Visual Representations.

---

### Meta-Review · Area_Chair_XTxQ · 2024-01-02

**Metareview:**

The paper proposes a new regularization term that can be added to the contrastive loss to improve the alignment of multimodal embeddings in settings with a limited amount of paired data. The regularization loss is designed to preserve the neighborhood structure of the unimodal embeddings.  As the authors state themselves, their contribution is to combine "two concepts that have not been studied together so far: (1) alignment of uni-modal pre-trained models and (2) a geometric regularization term aimed at preserving local neighborhood structures of the uni-modal models."

The proposed regularized alignment method is tested on speech-text and image-text alignment, showing improvements in the limited paired data regime over three baselines at the cost of computational overhead. However, the idea of using a geometry-persevering regularization has limited novelty, even so, it has not been tested on this task before.  This makes the paper a boarder-line submission

**Justification For Why Not Higher Score:**

The supposed regularization is new to the task but has generally limited novelty

**Justification For Why Not Lower Score:**

N/A

---

### Decision · Program_Chairs · 2024-01-16

Reject